# Three distinct developmental pathways for adaptive and two IFN-γ-producing γδ T subsets in adult thymus

Terkild Brink Buus [1], Niels Ødum [1], Carsten Geisler[1] & Jens Peter Holst Lauritsen[1]

Murine γδ T cells include subsets that are programmed for distinct effector functions during their development in the thymus. Under pathological conditions, different γδ T cell subsets can be protective or can exacerbate a disease. Here we show that CD117, CD200 and CD371, together with other markers, identify seven developmental stages of γδ T cells. These seven stages can be divided into three distinct developmental pathways that are enriched for different TCRδ repertoires and exhibit characteristic expression patterns associated with adaptive (γδTn), IFN-γ-producing (γδT1) and IFN-γ/IL-4-co-producing γδ T cells (γδNKT). Developmental progression towards both IFN-γ-producing subsets can be induced by TCR signalling, and each pathway results in thymic emigration at a different stage. Finally, we show that γδT1 cells are the predominating IFN-γ-producing subset developing in the adult thymus. Thus, this study maps out three distinct development pathways that result in the programming of γδTn, γδT1 and γδNKT cells.

[1] Department of Immunology and Microbiology, Faculty of Health and Medical Sciences, University of Copenhagen, 2200 Copenhagen, Denmark. Correspondence and requests for materials should be addressed to T.B.B. (email: terkild.buus@sund.ku.dk)

γδ T cells are a heterogeneous population with diverse effector functions during anti-microbial and anti-tumoural responses[1–3]. γδ T cells show great promise in anti-tumour immunotherapy[4]. However, while cytotoxic and IFN-γ-producing γδ T cell effector subsets elicit potent anti-cancer effects, other γδ T cell effector subsets have pro-oncogenic functions and are associated with poor prognoses[4, 5].

Unlike conventional αβ T cells, the effector functions of some γδ T cells are programmed during their development in the thymus[1]. The γδ T cell effector subsets can be divided based on their ability to produce either IL-17 (γδT17), IFN-γ (γδT1) or both IL-4 and IFN-γ (γδNKT)[1]. Whereas both of the IFN-γ-producing subsets γδT1 and γδNKT have been shown to be dependent on strong T cell receptor (TCR) signals during their development, γδT17 cells have been reported to develop in the absence of TCR ligand selection[6–9]. Additionally, studies have identified a population of γδ T cells that exhibit adaptive-like characteristics. Analogous to conventional αβ T cells, these adaptive γδ T cells are believed to be exported from the thymus as naïve (γδTn) cells that require peripheral priming for functional development, and can establish long-lasting TCR-dependent memory[2, 10–13]. While the development of γδTn cells is still largely undescribed, they have been suggested to develop in the absence of TCR ligand selection and to be exported with a naïve (CD62L$^+$CCR7$^+$CD44$^-$) surface phenotype[12, 14, 15].

The development of γδ T cells is initiated in the foetus and continues throughout life. Foetal and adult γδ T cell development may be considered two distinct systems that involve distinct progenitor waves[16, 17] and require specialised thymic micro-environments[6, 18], expressing distinct γδTCR repertoires and resulting in distinct effector subsets[18–21]. The dendritic epidermal T cell (DETC) subset, the natural γδT17 subset and a majority of the γδNKT subset develop only during foetal and perinatal life[18, 21, 22]. In adult mice, the effector subsets that develop are predominantly adaptive γδTn cells and IFN-γ-producing γδT1 and γδNKT cells, most of which utilise either the Vγ1.1 or the Vγ2 segment in their TCR[23] (V segment nomenclature as in ref. [24]).

αβ T cell progenitors can be divided into several distinct sub-populations based on their surface marker expression. These different subpopulations are correlated with distinct development checkpoints. By contrast, few surface markers have been identified on developing γδ T cells[25]. Most studies have solely used CD24 to distinguish immature (CD24$^{high}$) and mature (CD24$^{low}$) γδ thymocytes. Previous studies have further shown that CD25 marks a small population of highly immature γδTCR-expressing progenitors, and that CD73 marks γδ thymocytes that are committed to the γδ lineage[8, 26, 27].

Over the years, advances have been achieved in our understanding of how IL-17 vs. IFN-γ programming is determined in the thymus. This includes the identification of robust surface markers that distinguish IL-17 and IFN-γ-producing cells in the periphery and the perinatal thymus[6–9, 28–30]. However, in the adult thymus, where most of the γδ thymocytes are CD24$^{high}$, these markers primarily mark terminally differentiated or long-lived effector cells reminiscent of the perinatal stage, which are CD24$^{low}$[18,31,32]. These differences between γδ T cell development in the foetal and adult thymus and the scarcity of surface markers dividing the CD24$^{high}$ population, prompted us to identify additional surface markers to further segregate developing γδ T cells in the adult thymus.

In this study, we characterise CD117, CD200 and CD371 as surface markers that are expressed during γδ T cell development. Together with CD24, CD25 and CD73, these markers establish seven distinct development stages that are found in both the Vγ1.1$^+$ and Vγ2$^+$ subset. These seven stages can be divided into three pathways exhibiting different global gene transcription, including the expression of cytokines and transcription factors associated with γδTn, γδT1 and γδNKT cells. We show that cells within the three identified pathways display distinct TCRδ repertoires, and that progression through the pathways resulting in IFN-γ-producing effector cells can be induced by TCR signalling. Blocking thymic emigration causes an accumulation of γδ T cells at three stages representing the thymic end points of each pathway. The surface phenotypes of these end points indicate that γδTn and γδT1 cells are exported while still expressing CD24. Finally, although γδNKT cells are the major IFN-γ-producing subset in the thymus during homoeostasis, we show that γδT1 cells are the predominant IFN-γ-producing subset undergoing development in the adult thymus. Thus, this study maps out three distinct developmental pathways that result in programming of γδTn, γδT1 and γδNKT cells.

## Results

**Identification of additional markers of γδ development**. Studies of how and when different γδ thymocytes branch off and diverge towards different effector fates have been hampered by the scarcity of surface markers distinguishing different γδ thymocyte development stages in the adult thymus[25]. To identify additional development markers, we analysed our previously published data set, GSE75920[31], for genes encoding surface proteins that are differentially expressed by the CD24$^{high}$CD73$^-$ and CD24$^{high}$CD73$^+$ populations in both the Vγ1.1$^+$ and Vγ2$^+$ subsets (Fig. 1a, b and Supplementary Fig. 1a–c). Surface expression of the gene products was verified by flow cytometry yielding three markers that can further segregate CD24$^{high}$ γδ thymocytes: CD117 (c-Kit), CD200 (OX-2) and CD371 (Clec12a) (Fig. 1c–e).

**New markers allow isolation of seven distinct γδ populations**. Current developmental stages defined by CD24 (HSA), CD25 (IL-2Rα) and CD73 (Nt5e) are insufficient to delineate the development of γδ thymocytes towards different effector fates. To establish additional development stages, we examined the co-expression of CD117, CD200 and CD371 with CD24, CD25 and CD73. Unlike classical bi-axial flow cytometric plots, parameter reduction by t-distributed stochastic neighbour embedding (t-SNE) allowed visualisation of the co-expression of all six markers on individual cells. Figure 2a shows t-SNE-clustered cells coloured by their expression of each of the six markers. To identify distinct populations based on the expression of all six markers, we divided the cells into populations using automated clustering algorithms (Supplementary Fig. 2a). The best clustering was constructed by FlowSOM, dividing the total γδ thymocyte pool into seven distinct populations that are referred hereafter to as populations A through G (Fig. 2b, c). These populations were all present in mice from 1 to 28 weeks of age (Fig. 2d and Supplementary Fig. 2b). Furthermore, Vγ1.1$^+$ and Vγ2$^+$ progenitors were present in all seven populations, albeit with different distributions (Fig. 2e, f and Supplementary Fig. 2c). We could also gate each of the seven populations by conventional bi-axial flow cytometry gating (Fig. 2g). To determine how the identified populations relate to previously described γδ T cell populations, we investigated the expression of CCR6, CD5, CD27, CD44, CD45RB, CD122, CD127, NK1.1 and Ly-6C within each of the seven populations[7, 8, 10, 28, 29, 33]. We found that although they were differentially expressed, the existing markers were not sufficient to separate populations A to G (Fig. 2h–j and Supplementary Fig. 2d, e). Thus, co-expression of CD117, CD200 and CD371 together with CD24, CD25 and CD73 allowed the isolation of seven distinct γδ thymocyte populations.

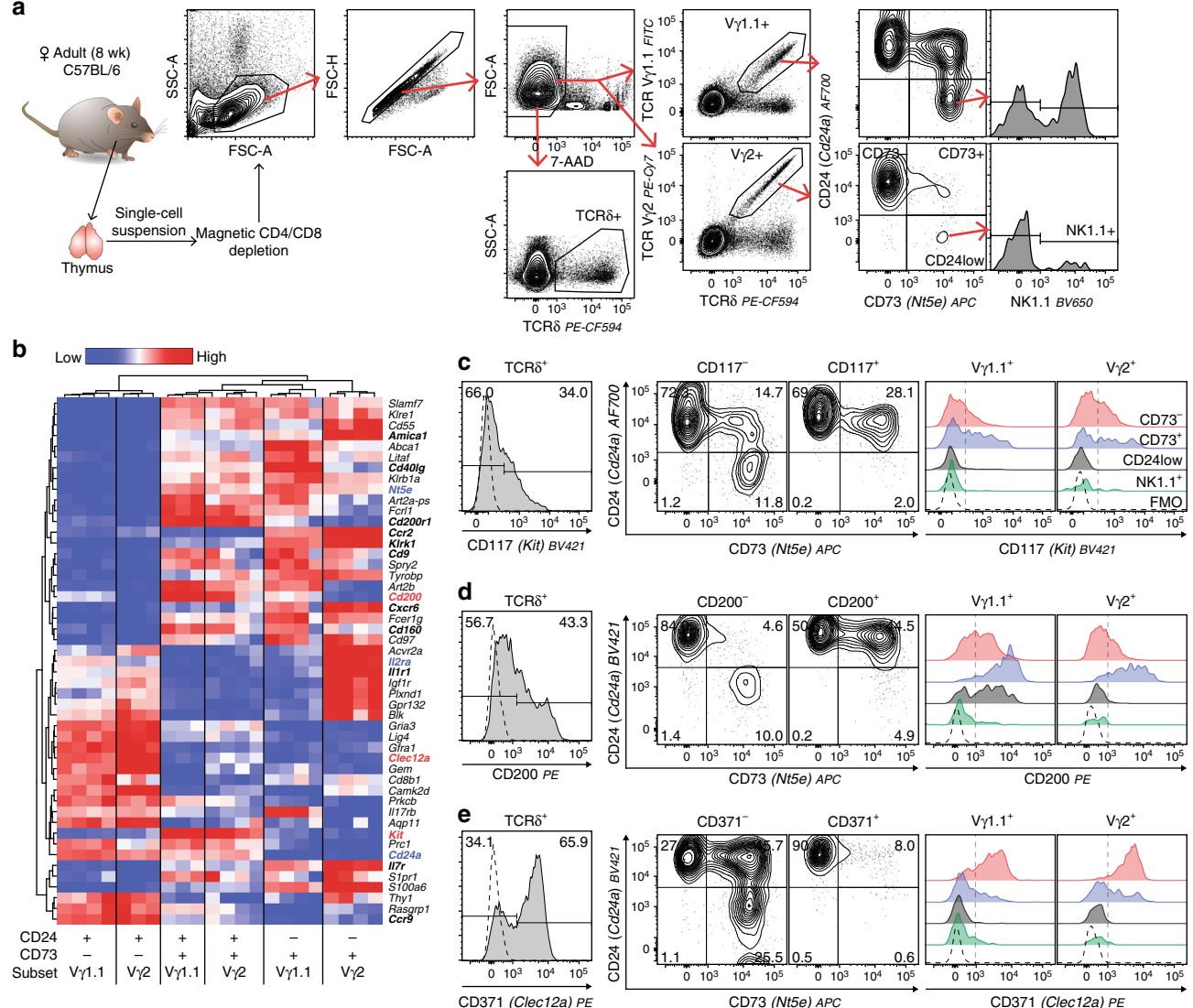

**Fig. 1** CD117, CD200 and CD371 are differentially expressed during γδ T cell development. **a** Gating strategy for analysis of surface makers within the total TCRδ⁺, the Vγ1.1⁺ and the Vγ2⁺ subsets. **b** Heat map of genes encoding surface-expressed proteins (GO:0009986 or GO:0005886) that were significantly differentially expressed between CD24⁺CD73⁻ and CD24⁺CD73⁺ cells. Data from public data set GSE75920 (three to four individual experiments from sorted cells, each pooled from 30 mice). Expression of genes marked in bold was verified by flow cytometry. Expression of red and blue genes is shown in **c–e**, and expression of remaining bold genes is shown in Supplementary Fig. 1. **c–e** Representative flow cytometric plots showing expression of **c** CD117, **d** CD200 and **e** CD371 on TCRδ⁺-gated thymocytes as histograms (left), distribution of marker negative and positive cells within development stages defined by CD73 and CD24 (centre) and expression of markers within development stages of the Vγ1.1⁺ and Vγ2⁺ subsets (right)

**Developing γδ T cells branch into effector populations**. Effector programming of γδ thymocytes is controlled at the transcriptional level, segregating the effector subsets with characteristic expression of distinct cytokines, signalling molecules and transcription factor networks[6, 17, 32, 34–36]. Consequently, we hypothesise that γδ thymocytes developing towards different effector fates cannot follow a direct sequential development but must branch into different development pathways as they diverge. To assess the relationship between the seven populations and effector programming, we sorted the Vγ1.1⁺ and Vγ2⁺ subsets of each population by flow cytometry (Supplementary Fig. 3a) and performed RNA-Seq using their total mRNA. This procedure enabled us to determine the transcriptional similarity of each population, and subsequently allowed us to link the transcriptional characteristics expressed in each population with their established functions in effector programming. We calculated the pairwise Euclidian distances between each of the populations as a measure of their individual transcriptional differences (Fig. 3a, b). By connecting the least transcriptionally different populations, we constructed models of how the populations were most likely to be related (Fig. 3c, d). Most populations showed high transcriptional similarity to more than one subsequent population, which indicated that progression through the A to G stages might not follow a sequential linear path but rather branch into different pathways as predicted in our hypothesis. This branching pattern was further supported by PCA analysis of the RNA-Seq data sets (Fig. 3e and Supplementary Fig. 3b–e), as well as by single-cell expression of CD24, CD25, CD73, CD117, CD200 and CD371 by flow cytometry analysed using two different algorithms that are used for inferring cellular progression: Isomap and Diffusionmap (Fig. 3f and Supplementary Fig. 3f)[37].

Subsequently, we investigated the expression of genes characteristic of γδ T cells programmed for distinct effector function (Fig. 3g). The expression of genes associated with TCR ligand

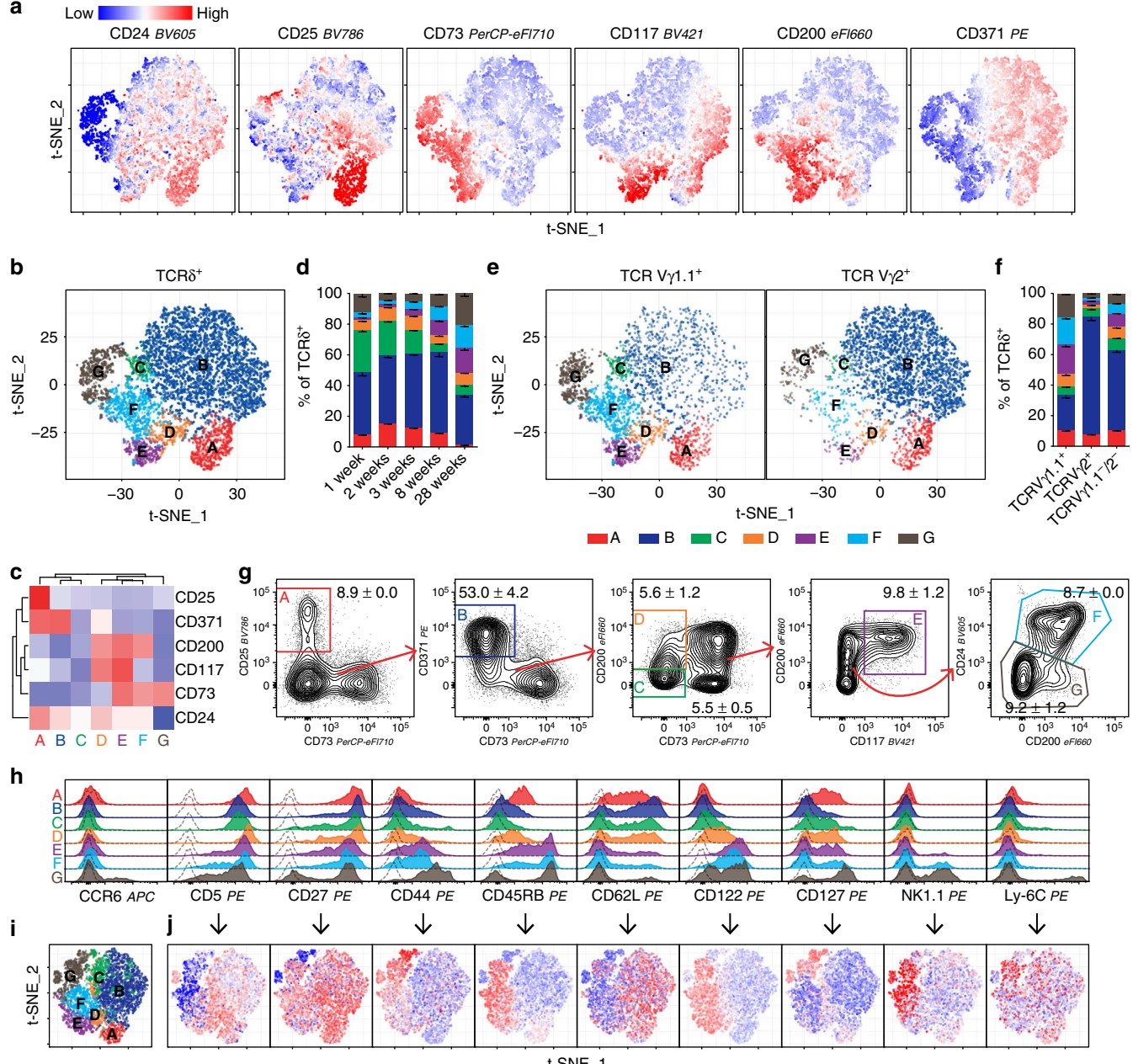

**Fig. 2** Surface expression of CD117, CD200 and CD371 allows the isolation of seven distinct γδ thymocyte populations **a**, **b** t-SNE map of TCRδ⁺ thymocytes coloured by **a** normalised expression intensities of CD24, CD25, CD73, CD117, CD200 and CD371 or by **b** FlowSOM automated clustering. Based on flow cytometric analysis of 2 × 5000 TCRδ⁺ cells pooled from two mice in silico. **c** Heat map of mean expression intensity of markers within the FlowSOM clusters named A to G. **d** Distribution of cells within populations A to G shown as a percent of the total TCRδ⁺ thymocytes from 1- to 28-week-old mice. Bars depict the mean ± SEM (n = 4, 4, 4, 2, 3 mice). **e**, **f** Distribution of cells within populations A to G within the Vγ1.1⁺ and Vγ2⁺ subsets of TCRδ⁺ thymocytes visualised as **e** t-SNE map coloured by FlowSOM clusters as in **b**, **f** stacked bar plots. Bars depict the mean ± SEM (n = 2 mice). **g** Standard bi-axial TCRδ⁺ gating of population A to G. Numbers denote the mean ± SEM of total TCRδ⁺ thymocytes (n = 2 mice). **h**, **i**, **j** Expression of established surface markers of γδ T cells within populations A to G visualised as **h** representative histograms or **i**, **j** representative t-SNE maps (n = 6 mice). **h** Dashed lines show background signal by FMO. **i**, **j** Colours depict **i** the location of events assigned to populations A to G within the t-SNE plot or **j** the normalised expression intensity of each marker

independent (γδT17 and γδTn) and TCR ligand-dependent (γδT1 and γδNKT) programming clearly segregated the A, B and C populations from the E, F and G populations, respectively. Remarkably, the E and G populations each expressed distinct gene sets associated with γδT1 and γδNKT cells, respectively (Fig. 3g). While most of the genes related to TCR ligand independent γδT17 and γδTn programming were highly expressed by the B and C populations, a subset of these genes,

including the IL-17A and F genes themselves, were primarily expressed by the Vγ2⁺ G population, which is known to contain the majority of thymic resident γδT17 cells, reminiscent of perinatal development[32]. To verify the RNA-Seq expression at the protein level, we analysed the expression of RORγ(t) (encoded by *Rorc*), PLZF (encoded by *Zbtb16*) and AHR within the populations by flow cytometry, which confirmed the segregation

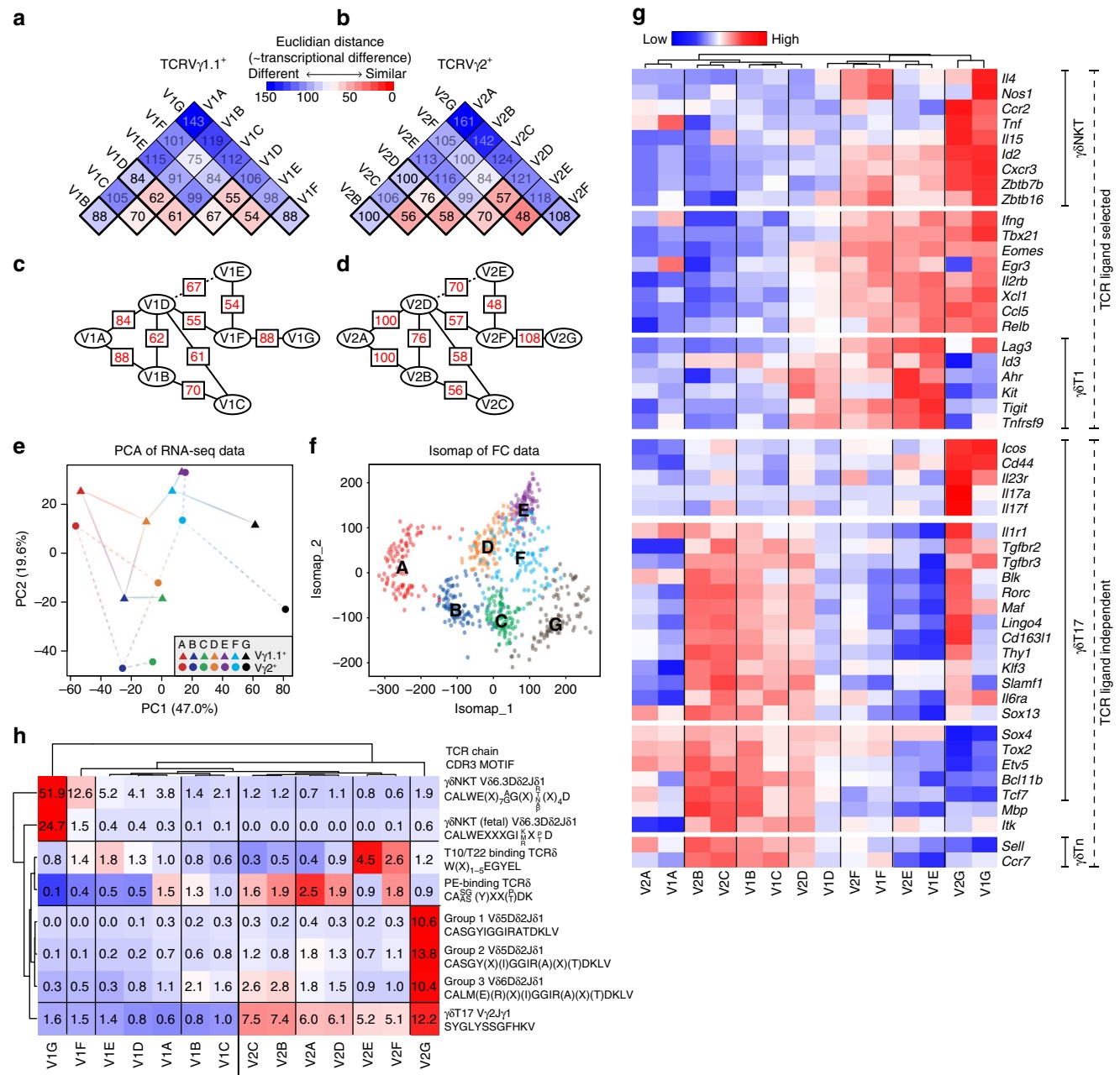

**Fig. 3** Developing γδ T cells branch into populations with distinct effector characteristics **a**, **b** Pairwise Euclidian distances between each of the A to G populations calculated from whole transcriptome RNA-Seq expressions within **a** the Vγ1.1+ (V1A–V1G) and **b** the Vγ2+ (V2A–V2G) subset. Numbers and colours denote the mean Euclidian distances. **c**, **d** Diagrams of the shortest distances between the A to G populations, associating the most transcriptionally similar populations by lines within **c** the Vγ1.1+ and **d** the Vγ2+ subset. Red numbers denote the Euclidian distance between the two connected populations. **e** Principal component analysis (PCA) of RNA-Seq expression within the A to G populations of the Vγ1.1+ and Vγ2+ subsets. Lines correspond to the lowest pairwise distances shown in **c**, **d**. **f** Cellular progression predicted by flow cytometric expression of CD24, CD25, CD73, CD117, CD200 and CD371 by Isomap. Each A to G populations was reduced to 100 cells before analysis. Each dot represents a single cell. **g** Heat map showing expression of key transcription factors and effector molecules related to different γδ T cell effector subsets within the A to G populations of the Vγ1.1+ (V1A–V1G) and the Vγ2+ (V2A–V2G) subset. **h** Abundance of known TCRγ and TCRδ CDR3 motifs associated with different γδ effector subsets within the A to G populations of the Vγ1.1+ and the Vγ2+ subset. Numbers and colour depict the mean percentage of RNA-Seq reads including the indicated CDR3 motifs. Gene expression data are analysed as the mean expression from two independent RNA-Seq libraries, each constructed from RNA isolated from 24 mice across three independent cell sorting runs

of effector-related proteins between populations (Supplementary Fig. 3g–i).

In addition to their expression of distinct transcriptional networks, different γδ T cell effector subsets are enriched for certain TCR sequences. Thus, we next determined if the segregation of the populations into effector subsets was

accompanied by the corresponding CDR3 motifs (Fig. 3h). The consensus γδNKT CDR3δ sequence was represented in more than half of the TCRδ transcripts within the Vγ1.1+ G population. CDR3 motifs related to γδT17 cells were all highly enriched within the Vγ2+ G population[19, 20]. The T10/T22-binding motif was highly enriched within the E population from

both the Vγ1.1[+] and the Vγ2[+] subset. T10/T22-reactive γδ T cells constitute up to 1% of the γδ thymocyte pool and are prototypical γδT1 cells[7]. Finally, the germline Vγ2Jγ1 motif associated with γδT17 cells and a PE-binding CDR3δ motif associated with the γδTn cell subset[12] were both selectively reduced within the E and F populations in concordance with γδT17 and γδTn cells being programmed in the absence of TCR ligand selection (Fig. 3h).

Together, these phenomena show that developing γδ T cells branch into populations with distinct effector characteristics. Based on these data, we propose a model of γδ T cell development in which γδ T cell progenitors from both the Vγ1.1[+] and Vγ2[+] subset encounter multiple checkpoints during their development that direct them to progress through distinct pathways. Based on evidence from previous studies[6, 7, 9, 29], the selection of TCR ligands or the lack thereof is likely to constitute at least one of these checkpoints. Finally, we hypothesise that progression through these different pathways facilitates the programming and export of γδ T cells with distinct effector functions.

**TCR signalling induces progression through the D–F stages.** TCR selection is known to direct effector programming towards the IFN-γ-producing subsets[6, 7, 9, 29]. Accordingly, if the E and G populations are indeed thymic precursors of γδT1 and γδNKT cells, respectively, TCR stimulation should direct γδ thymocytes to progress through the pathways resulting in these populations. These pathways are marked by initial CD200 induction (population D) followed by CD73 expression with or without CD117 expression (populations E or F) and eventually downregulation of CD24 (population G). To test this phenomenon, we sorted cells from each of the A to G populations and cultured them for 2 days on OP9-DL1 monolayers in the presence or absence of anti-CD3 stimulation. While CD25 marks the earliest TCRδ[+] cells in the thymus[8, 26], it is also known to be induced on most T cells after anti-CD3 stimulation in vitro. Indeed, CD25 was induced by anti-CD3 stimulation in all samples, irrespective of which population that was initially sorted, thus obscuring the distinction of true population A cells (Fig. 4a, b). To avoid faulty classification, we merged the A and B populations in the analysis, both defined by being CD371[+]CD73[−] (Supplementary Fig. 4a). In cells sorted from the A, B and C populations, TCR stimulation induced expression of CD200, CD117 and CD73 (Fig. 4a, b). In contrast, cells sorted from the D, E, F and G stages were largely unaffected by TCR stimulation (Fig. 4c and Supplementary Fig. 4b, c). This result suggests that progression through the CD200-expressing (D, E and F) stages is induced by TCR signalling, in agreement with these cells being precursors of IFN-γ-programmed effectors. Concordantly, the D, E and F populations expressed high levels of TCR ligand-induced genes (Fig. 4d)[9]. The G populations were unaffected by TCR signalling and did not express hallmarks of recent TCR signalling ex vivo (Fig. 4a–d).

We hypothesised that populations undergoing thymic TCR ligand selection would exhibit different TCR repertoires than would cells developing in the absence thereof. According to their transcriptional profiles and the TCR inducibility of CD200, the D, E and F populations would be expected to express different TCR sequences than those found in the B and C populations. To assess this scenario, we analysed the expression of TCR Vδ within the populations of the RNA-Seq data set. We found that the D, E and F populations were highly enriched for certain TCR Vδ sequences (*Trav21/dv12*, *Trav15-1/dv6-1*, *Trdv5* and *Trav13-4/dv7*) compared with the B and C populations (Fig. 4e). These data are all consistent with the D, E and F populations being TCR ligand-selected precursors of IFN-γ-programmed effectors.

**γδ T cells emigrate from the thymus at three stages.** The gene expression profiles suggest that E and G are the terminal populations in the γδT1 and γδNKT pathways, respectively, and should be ready to emigrate from the thymus. Furthermore, as γδTn cells are believed to develop in the absence of TCR ligand selection, we hypothesised that either B or C could be the terminal population for these pathways. To test these hypotheses, we blocked S1P1-mediated thymic export by treating mice with the S1P1 antagonist FTY720[38]. Blocking thymic export should result in thymic accumulation of cells within the B/C, E and G populations and their subsequent depletion from the peripheral lymph nodes. In mice treated with FTY720, we observed an accumulation of cells within the C, D, E, F and G populations after 5 days of treatment (Fig. 5a, b and Supplementary Fig. 5a, b). From day 5 to day 10, the D and F populations increased only slightly, whereas the C, E and G populations continued to accumulate and increased to levels that were ten-, five- and tenfold higher than in the untreated mice, respectively. Concurrent with their accumulation, after FTY720 treatment the C and E populations displayed downregulated CD24 levels comparable to mature γδ T cells in the periphery (Fig. 5c and Supplementary Fig. 5c). While most peripheral γδ T cells express low levels of CD24, a large proportion of the γδ T cell recent thymic emigrants (γδRTEs) are CD24[high] when they reach their target organs, upon which they downregulate their CD24 expression[39]. The presence of CD24[low] cells within the C and E populations, solely when thymic export was blocked, suggested that at homoeostasis these populations are exported while still expressing high levels of CD24. To assess whether the accumulation of cells within the C and E populations causes subsequent depletion of CD24[high] γδRTEs in the periphery, we analysed the expression of CD24 and CD73 on peripheral γδ T cells in inguinal lymph nodes (iLN) after FTY720 treatment. In untreated mice, we found CD24[high] γδRTEs within both CD73[−]- and CD73[+]-gated cells corresponding to the expression pattern of the thymic C and E populations, respectively (Fig. 5d, e and Supplementary Fig. 5d). Both CD24[high] γδRTEs populations were significantly depleted in the iLN when thymic emigration was inhibited. These results show that γδ thymocytes accumulate at the C, E and G stages in the thymus and are subsequently depleted from the periphery when thymic emigration is inhibited. This phenomenon is consistent with the cells from the C, E and G populations constituting terminal populations of three distinct pathways that lead to the export of γδTn cells, γδT1 cells and γδNKT cells, respectively.

**The C population exhibits hallmarks of γδTn cells.** Both γδT17 and γδTn cells are believed to develop in the absence of TCR ligand selection[6, 7, 9, 14, 29]. As γδT17 cells are restricted to develop only during perinatal life[18], we hypothesised that the cells developing through the C pathway in adult mice represent the γδTn cell subset. In this case, the number of IL-17A-producing γδ T cells should remain unchanged, despite the accumulation of the C population after FTY720 treatment. Thus, we analysed the expression of IL-17A and IFN-γ within the accumulating γδ thymocytes after FTY720 treatment and found that while the number of IFN-γ[+] γδ T cells was increased 14-fold, the number of IL-17A[+] γδ T cells was unchanged (Fig. 6a–c and Supplementary Fig. 6a, b). Furthermore, the C populations expressed high levels of *Ccr7* and *Sell* (encoding CD62L) and low levels of *Cd44*. This *Ccr7*[high]*Sell*[high]*Cd44*[low] phenotype is associated with γδTn cells and contrasts the memory-like *Cd44*[high] γδT17 phenotype (Fig. 3g). Consequently, despite their independence of TCR ligand selection, their expression of γδT17-related genes and their enrichment for γδT17-related CDR3 motifs, the cells from the C populations did not produce IL-17A and expressed a naive

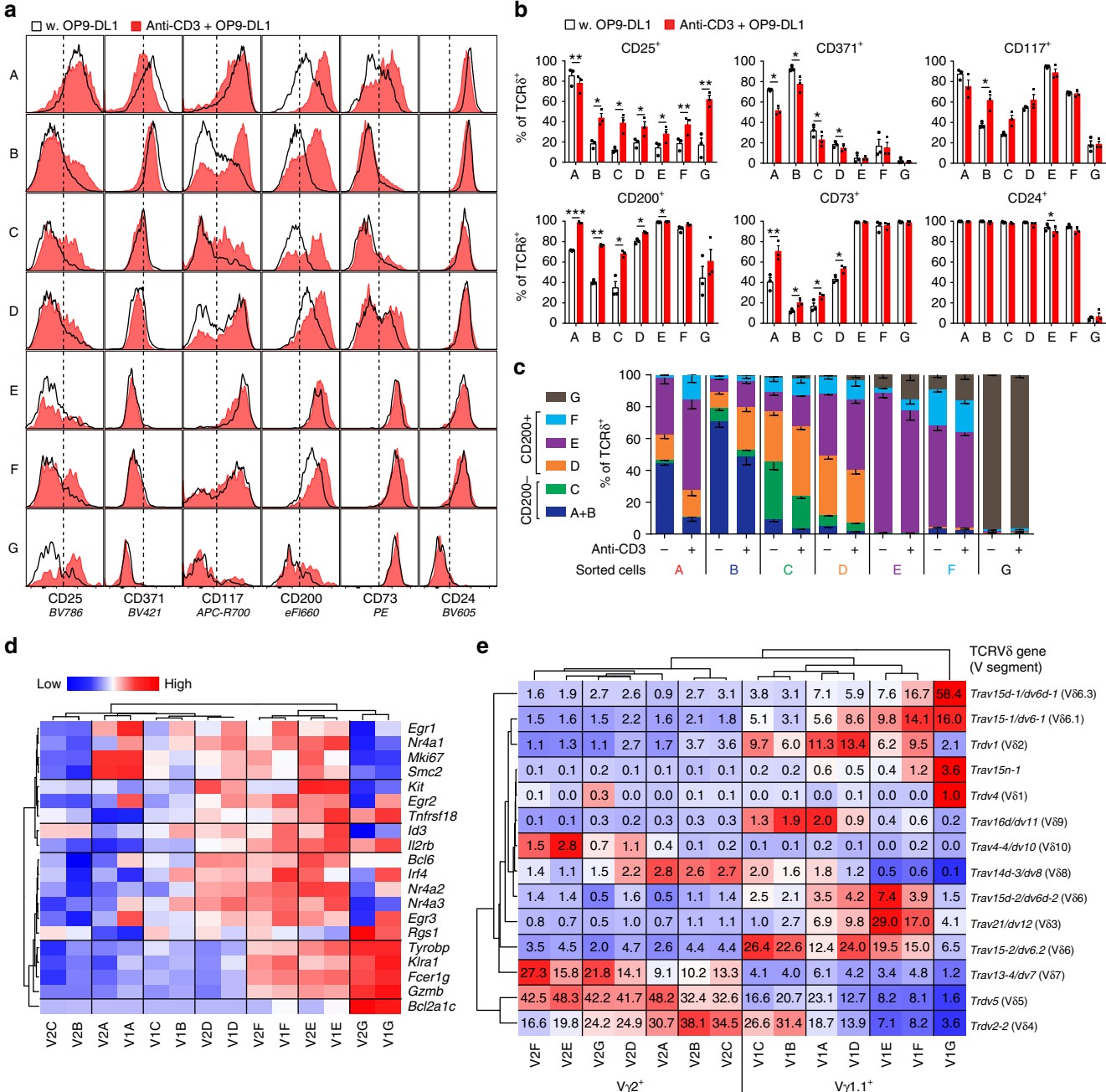

**Fig. 4** Progression through the D, E and F populations is induced by TCR signalling. **a–c** Changes in surface marker expression of TCRδ⁺ cells from sorted population A to G cells after 2 days of culture on OP9-DL1 monolayers in the presence or absence of immobilised anti-CD3. The data are visualised as **a** histograms and **b** bar plots of the percentage of cells positive for each individual marker as well as **c** gated into the A to G populations. Bars depict the mean ± SEM from three independent experiments with cells sorted from four to eight mice. Statistical analyses were performed using the paired two-sided Student's $t$ test, with significance defined as $*p < 0.05$; $**p < 0.01$; $***p < 0.001$. **d** Heat map showing expression of genes reported to be induced by γδTCR signalling during development within the A to G populations of the Vγ1.1⁺ (V1A–V1G) and the Vγ2⁺ (V2A–V2G) subset. **e** TCRδ repertoire of cells within the A to G populations of the Vγ1.1⁺ and the Vγ2⁺ subset. Numbers and colour depict the mean percentage of RNA-Seq reads aligned to the TCRδ locus from two independent RNA-Seq libraries, each constructed from RNA isolated from 24 mice across three independent cell sorting runs

surface phenotype. This finding indicates that the cells in the C population are likely progenitors of γδTn cells.

**γδT1 are the predominant developing IFN-γ-producing subset.** Unlike IL-17A⁺ γδ T cells, IFN-γ⁺ γδ T cells drastically accumulated in the adult thymus after FTY720 treatment (Fig. 6a–c). As IFN-γ is produced by both γδT1 and γδNKT cells, we next wanted to determine if the accumulating IFN-γ⁺ cells belong to

the γδT1 or γδNKT subset. Thus, we co-stained the accumulating IFN-γ⁺ cells with the γδNKT markers PLZF and NK1.1, as well as CD45RB, which has been associated with γδT1 cells[28, 40, 41]. While few of the accumulating IFN-γ⁺ cells co-expressed NK1.1⁺ and PLZF⁺, most of the accumulating IFN-γ⁺ cells were CD45RB^hi, NK1.1⁻ and PLZF⁻ (Fig. 6d, e). Consequently, a larger proportion of the cells from the F populations must progress to the E stage and become γδT1 cells rather than γδNKT cells. This possibility is further supported by the higher similarity between

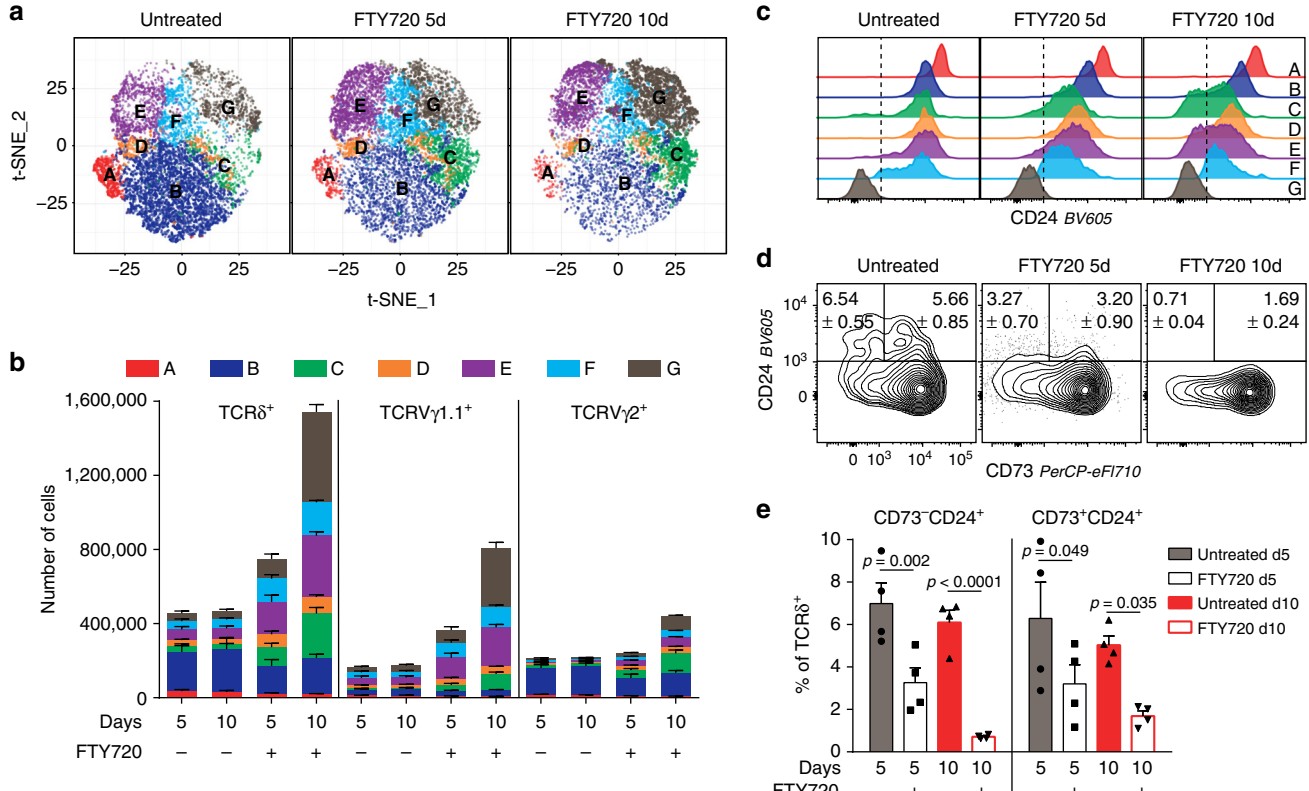

**Fig. 5** γδ T cells emigrate from the thymus at the C, E and G stages **a**, **b** TCRδ⁺ thymocyte distributions within population A to G after treatment with FTY720 for 5 (5d) or 10 days (10d) visualised using **a** t-SNE maps and **b** stacked bar plots of cell numbers within TCRδ⁺, Vγ1.1⁺ and Vγ2⁺ subsets. **c** Representative CD24 expression within A to G populations after treatment with FTY720 for 5 or 10 days. **d**, **e** Expression of CD24 within CD73⁻ and CD73⁺ TCRδ⁺ cells from inguinal lymph nodes after FTY720 treatment for 5 and 10 days as **d** representative flow cytometry plots and **e** quantitative bar plots showing the percent of TCRδ⁺ cells within each population. Data from two independent experiments for both 5- and 10-day treatments ($n = 4$ mice per group per time point). Numbers within gates and bar plots depict the mean ± SEM. Statistical analyses were performed using the unpaired two-sided Student's $t$ test

the E and F populations than between the F and G populations, at the surface marker level, at the transcriptional level and at the TCR repertoire level (Figs. 3 and 4e). Consequently, while γδNKT cells within the G population constitute the majority of the thymic resident IFN-γ-producing γδ T cells, they are not a major actively developing effector subset in the adult thymus. Consistent with the G populations primarily containing long-lived resident effector cells at homoeostasis, the G populations showed the lowest TCR diversity and were largely oligo-clonal (Supplementary Fig. 7a–d). The higher propensity of F population cells to develop towards the γδT1 subset rather than γδNKT subset is further emphasised by the reduction in Vδ6.3⁺ frequency within the Vγ1.1⁺ G population after FTY720 treatment (Fig. 6f, g). In fact, while the A through F populations maintained similar Vδ6.3⁺ and Vδ4⁺ frequencies after FTY720 treatment, this was not the case for the G populations (Fig. 6f–i). Taken together, these findings show that γδT1 cells are the predominant IFN-γ-producing subset developing in the adult thymus.

**Discussion**

Studies of how and when different γδ thymocytes branch off and diverge towards different effector fates in the adult thymus have been hampered by the scarcity of surface markers distinguishing different γδ thymocyte developmental stages[25]. In this study, we characterise three new development markers, CD117, CD200 and CD371, allowing the identification of seven γδ thymocyte populations within both the Vγ1.1⁺ and the Vγ2⁺ subset. We used this improved resolution to establish three developmental pathways

that lead to the development of distinct γδ T cell effector subsets exhibiting hallmarks of IFN-γ-producing (γδT1), IFN-γ/IL-4-co-producing (γδNKT) and adaptive (γδTn) γδ T cells (Fig. 7).

Several surface markers expressed by different γδ T cell effector populations are well established. However, while these markers are invaluable for distinguishing effector populations in the periphery and foetal thymus, they do not clearly distinguish developing γδ thymocytes in the adult thymus. In fact, many of these markers primarily segregate the effector cells contained within the G populations, many of which appear to be long-lived resident effector cells. CD44, CD45RB and CD122 are upregulated in the E, F and G populations believed to be selected by the TCR in a pattern similar to CD73. However, neither CD44, CD45RB nor CD122 could separate the populations as clearly as CD73. Furthermore, the existing markers were unable to distinguish the C and D populations from the B population in the absence of CD200 and CD371, and could not segregate the E from the F population without the detection of CD117.

Due to the lack of development markers and a focus on the foetal system, most previous studies have focused solely on CD24^low cells as the thymic end point of γδ T cell development[32, 42, 43]. This is despite strong and long-standing evidence of CD24 expression within the majority of γδ T cell recent thymic emigrants (γδRTEs) in adult mice[39, 44]. These CD24^high γδRTEs have been shown to constitute approximately half of the total γδ T cell populations in peripheral lymphoid organs and to be continuously replenished[39]. In this study, we identified two populations of CD24^high γδ thymocytes, stages C and E, which

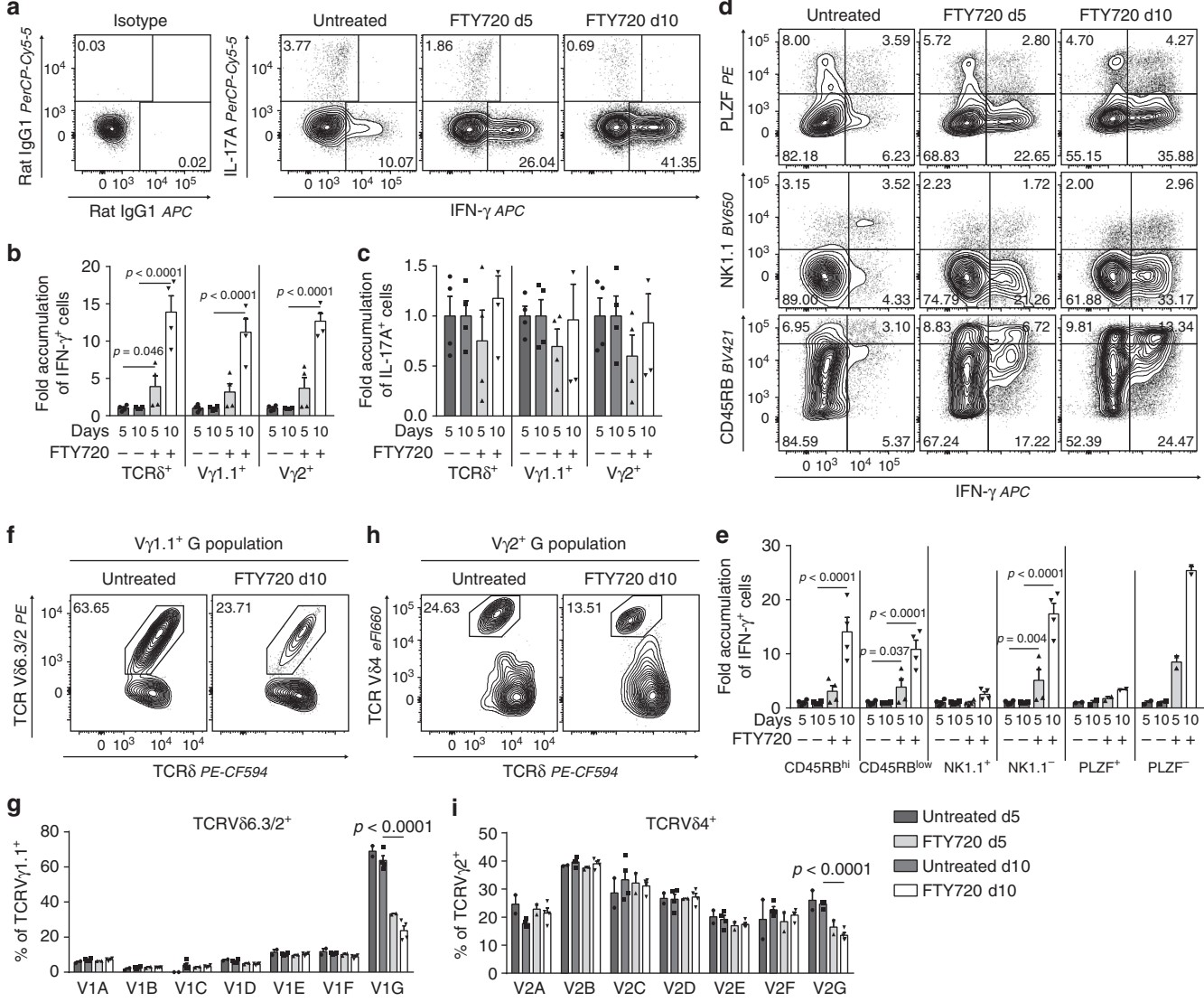

**Fig. 6** γδT1 cells, but not IL-17A⁺ γδ T cells, accumulate when thymic emigration is inhibited. **a–c** Expression of IFN-γ and IL-17A within TCRδ⁺ cells after treatment with FTY720 for 5 (5d) or 10 days (10d). **a** Representative flow cytometry contour plots and normalised quantification of the fold change in **b** IFN-γ⁺ and **c** IL-17A⁺ TCRδ⁺ cells (n = 4). **d, e** Co-expression of IFN-γ with PLZF (n = 2), NK1.1 (n = 4) or CD45RB (n = 4) within TCRδ⁺ cells after treatment with FTY720 for 5 or 10 days. **d** Flow cytometry plots and **e** normalised quantification of the fold change within TCRδ⁺IFN-γ⁺ cells. **f–i** Percentage of cells within the A to G populations expressing **f, g** Vδ6.3/2 within the Vγ1.1⁺ subset and **h, i** Vδ4 within the Vγ2⁺ subset after treatment with FTY720 for 5 (n = 2) or 10 (n = 4) days. Bars depict the mean ± SEM from two independent experiments for both 5 and 10 days (n = 2–4 mice per group per time point). Statistical analyses were performed using the unpaired two-sided Student's t test

accumulated when thymic export was blocked. Additionally, we found that these accumulated populations downregulated CD24 only when their export was pharmacologically blocked. This result is analogous to γδRTEs, which are known to downregulate CD24 after they arrive at their target organs in the periphery[39]. Concurrent with the thymic accumulation of cells within the C and E stages, we observed selective depletion of CD24^high γδ T cells in the peripheral lymph nodes after FTY720 treatment. Consistent with thymic export at both the (CD73⁻) C and (CD73⁺) E stages, the CD24^high γδRTEs were equally distributed between CD73⁻ and CD73⁺ cells in the peripheral lymph nodes. Taken together, these findings support the occurrence of thymic export at the C and E stages and establish a gating strategy that will allow future studies to discriminate immature developing CD24^high cells from cells that exit the thymus while still expressing CD24.

Effector programming of γδ T cells is coupled to TCR signals received during their development[6, 7, 9]. While the ligands for most γδTCRs are still unknown, sequencing technologies have increased the number of known complementarity-determining region three (CDR3) motifs associated with γδT1, γδT17 and γδNKT cells[19–21, 45, 46]. Here we found that TCR signalling directed development to proceed through the D, F and E stages, and we showed that the TCR repertoires were highly different between the TCR ligand-dependent and -independent stages. Furthermore, we showed that known sequence motifs of prototypical TCR ligand-dependent T10/T22-reactive γδT1 cells were enriched within the D, F and E population, with a concomitant depletion of motifs related to TCR signalling naive γδT17 and γδTn cells[7, 19, 20]. The E stages expressed the highest levels of Id3, Lag3 and Tigit, as well as the lowest levels of Sox13, Bcl11b and Thy1. This profile has been shown to be induced by TCR ligand selection of γδ intraepithelial lymphocytes (γδIELs)

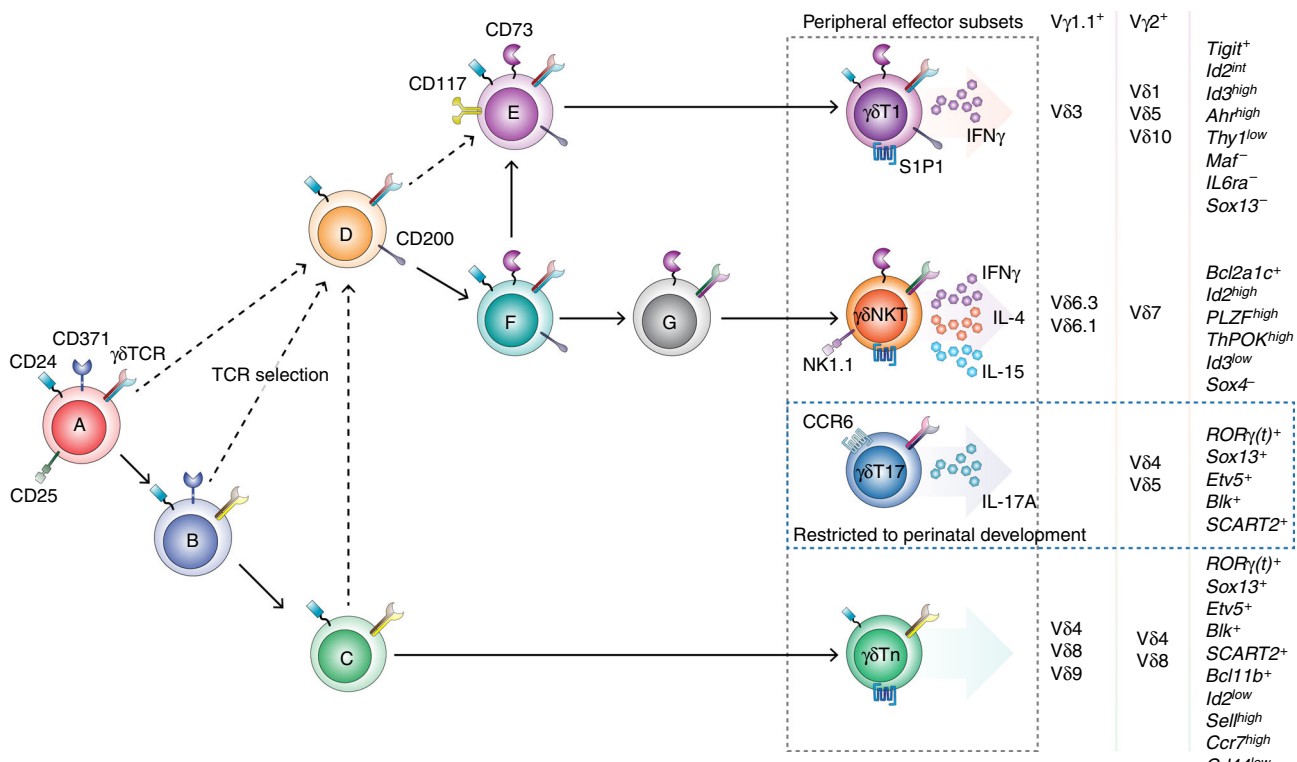

**Fig. 7** γδTn, γδT1 and γδNKT cells develop through three distinct pathways in the adult thymus. Suggested model of γδ T cell development divided into the A to G stages defined by the expression of CD117, CD200 and CD371 together with CD24, CD25 and CD73. The A stage defines the earliest γδTCR-expressing progenitors, which develop towards the B and C populations and eventually result in the export of TCR-naive adaptive γδ T cells (γδTn). Encounter with cognate TCR ligands induces TCR selection, shifting the cells to progress through the D and F stages. TCR-selected γδ thymocytes will then progress to the E or G stage, resulting in the export of IFN-γ-producing (γδT1) or IFN-γ/IL-4-co-producing (γδNKT) cells, respectively. The C, E and G stages are characterised by the distinct expression of key factors involved in effector programming, including characteristic transcription factor networks and cytokines, and display highly focused TCR Vδ repertoires. IL-17-producing (γδT17) cells do not develop in the adult thymus, but long-lived resident γδT17 cells are included in the G population at homoeostasis and can be distinguished by their expression of CCR6

in the gut and skin[6, 47]. Intriguingly, the E stages also express high levels of *Ahr*, which, together with CD117, is expressed by both epidermal and gut IFN-γ-producing γδIELs (GSE85422[47]; Imm-Gen.org; Kadow, Jux[48]). Similar to the T10/T22-reactive γδT1 cells, these γδIEL populations are selected by known ligands. This supports that the upregulation of CD117 is a common trait for γδT1 cells[6, 7, 9]. Unlike the IFN-γ-producing subsets, it has been hypothesised that γδT17 cells and γδTn cells develop and are exported in the absence of TCR selection by endogenous ligands[6, 7, 10, 14]. When thymic export was blocked, we observed an accumulation of cells within the C population with an apparent TCR ligand-naive phenotype expressing genes associated with γδTn and γδT17 programming. However, despite having the associated transcriptional networks[32, 35], these cells did not produce IL-17A upon stimulation. Unlike γδT17 cells, the C population exhibited hallmarks of γδTn cells (*Ccr7*high*Sell*high*Cd44*low), further indicating that the C population contains mature γδTn cells. Consequently, our data suggest that similar transcriptional networks are active in the development of γδTn and γδT17 cells and support that expression of these transcriptional networks are insufficient to allow IL-17A programming in the adult thymus[18, 22]. The G population contains a mixture of γδNKT and γδT17 cells that are enriched within the Vγ1.1+ and Vγ2+ subset, respectively, as evidenced by their transcriptional profiles and the expression of CD27, CD44, CCR6 and NK1.1 associated with these effector populations[8, 28]. We found that only the G populations had an altered proportion of Vδ6.3/2+ and Vδ4+ cells after FTY720 treatment and were highly enriched

or oligo-clonal for semi-invariant TCR sequences at homoeostasis. Furthermore, while IFN-γ-producing cells readily accumulated in the adult thymus, we observed no accumulation of IL-17A-producing cells. This finding suggests that, at homoeostasis, the G populations contain a large proportion of long-lived resident effector cells that have not recently undergone thymic development. This phenomenon is consistent with the restriction or bias of γδT17 and γδNKT cells towards development during the perinatal stage, respectively[18, 21]. Consequently, studies should be careful when comparing actively developing γδ thymocytes with resident effector γδ T cells included in the G population, and we strongly recommend including CCR6 and NK1.1 to further distinguish the two effector subsets contained in the G population. However, unlike γδT17 cells, γδNKT cells do develop in the adult thymus, and thus we have denoted the actively developing part of the G population as the thymic precursors of γδNKT cells in our proposed developmental model (Fig. 7).

The development of γδNKT cells is known to be dependent on strong TCR signalling required for induction of ThPOK and PLZF[41, 42, 49, 50]. The mechanism that determines whether a cell progresses towards the γδT1 or γδNKT effector fate, both of which are dependent on strong TCR signals during development, remains poorly defined. Differential dependence on Id3 and SLAM/SAP signalling as well as distinct progenitor origins have been suggested[41, 51, 52]. A previous study has also shown that the development of γδNKT cells is dependent on TCR signalling induced by outside-in conformational changes, whereas other

IFN-γ-producing γδ T cells are not[53]. Interestingly, the hallmarks of γδNKT cells, including the expression of *Il4*, *Zbtb7b* (encoding PLZF) and *Zbtb16* (encoding ThPOK) as well as TCRVδ6.3, were all specifically enriched in the Vγ1.1+ G population as well as a small fraction of the F population, but not within the E population. Thus, we have shown that TCR-selected γδ thymocytes progressing towards γδNKT or γδT1 cells can be distinguished by their differential expression of CD117. Interestingly, this distinction between γδNKT and γδT1 is emphasised in our results by a dichotomy in the expression of *Id2* and *Id3* between G and E stages that are known to be involved in the development of these subsets[41, 54]. When thymic export was blocked, we found a drastic accumulation of IFN-γ-producing γδ T cells. Remarkably, most of these cells were negative for PLZF and NK1.1 and expressed high levels of CD45RB, consistent with their status as γδT1 cells. This finding indicates that homoeostatic development in adult mice favours the γδT1 over the γδNKT effector fate, yet we cannot exclude the possibility that some IFN-γ is produced by γδTn cells that are activated by short-term stimulation, as shown for peripheral CD27+ γδ T cells[8]. The bias towards γδT1 development in adult mice may seem contradictory to the γδNKT phenotype dominating the mature (CD24low) population in the thymus at homoeostasis. However, within the Vγ1.1+ G population, we found high proportions of germline-like Vδ6.3 CDR3 sequences that were characteristic of γδNKT cells developing during the perinatal period, where TdT is not expressed[21]. Furthermore, we showed that the proportion of cells expressing the prototypical Vγ1.1Vδ6.3+ within the G population was markedly diluted by other TCRVδ-bearing Vγ1.1+ cells when thymic export was blocked. A similar trend was observed in the resident γδT17 population dominating the Vγ2+ G population at homoeostasis. In this population, we observed marked enrichment in germline CDR3 sequences associated with γδT17 cells, which are known to develop during the perinatal period[18–20]. This finding indicates that the G populations, at homoeostasis, contain long-lived resident γδ T effector cells that developed during the perinatal period, as observed in many other immune and peripheral organs[18].

In conclusion, this study presents three new markers that allow the identification of seven distinct developmental stages of γδ T cells. We provide data supporting that these stages branch into three pathways that result in the programming and export of γδTn, γδT1 and γδNKT cells. The functions of these effector subsets are highly specialised and provide immunity against various pathogens and transformed host cells. Thus, we present a framework for investigating the processes that precede effector programming. We believe that this framework will allow future studies to gain further insight and determine how we can shape and utilise our γδ T cell pool towards the most beneficial effector responses.

## Methods

**Mice and FTY720 treatment**. Unless stated otherwise, all mice experiments were conducted using 7- to 8-week-old female C57BL/6 mice obtained from Taconic (B6-F; Ry, Denmark) and housed in specific pathogen-free facilities at the Department of Experimental Medicine, Faculty of Health and Medical Sciences, University of Copenhagen, in accordance with the national animal-protection guidelines (approved by the Danish Veterinary and Food Administration; license no. 2012-15-2934-00663). To analyse 1-, 2- and 3-week-old mice, litters with mixed genders were used. Mice were killed by careful cervical dislocation to avoid bleeding into the thymus. Standard deviation was established in pilot experiments. We used two to four mice for phenotypic characterisation.

In the FTY720 experiments, mice were randomly assigned to treatment groups, and 2.5 µg per mL FTY720 (Sigma-Aldrich) was added to the drinking water of treated cages (containing two to four mice each). For both treated and untreated mice, cages and drinking water were replaced after 5 days, and new FTY720 was added to the treated cages. Groups of four mice per group per time point were used. No mice were excluded from the analysis. The analysis was performed without blinding.

**γδ thymocyte co-culture with OP9-DL1 cells**. For short-term culture of sorted γδ thymocytes, OP9-DL1 feeder cells were used. OP9-DL1 cells (gift from J.C. Zúñiga-Pflücker[55]) were maintained at 30–80% confluency in OP9 medium (HEPES-buffered Glutamax MEM-α (Life Technologies) containing 20% FBS, penicillin/streptomycin, Na-pyruvate, gentamycin and β-mercaptoethanol) in 10 cc culture dishes (Nunc, Roskilde, Denmark). On the day before the experiments, OP9-DL1 cultures were set up by adding $3 \times 10^3$ OP9-DL1 cells per well of a flat-bottom 96-well plate (Nunc, Roskilde, Denmark) and allowed to form monolayers overnight. For anti-CD3-treated cultures, the wells were coated with anti-CD3 prior to seeding OP9-DL1 cells by adding 5 µg per mL anti-CD3 antibody (145-2C11 clone; BioLegend) in PBS to each well and incubated for at least 4 h at 37 °C. The medium was removed, and $2–10 \times 10^3$ sorted γδ thymocytes were suspended in 200 µL OP9 medium containing 5 ng per mL rmIL-7, and 5 ng per mL rhFlt3-L (R&D Systems Inc., Minneapolis, USA) and added to each well. OP9-DL1 co-cultures are known to induce artificial phenomena such as fate switching[27, 31, 56]. To avoid these phenomena and limit the risk of selective outgrowth of minor populations, the cells were only cultured for 2 days before analysis by flow cytometry.

**Identification of surface marker candidates**. MicroArray data from CD24+ CD73−, CD24+CD73+ and CD24−CD73+ Vγ1.1+ and Vγ2+ thymocytes were acquired through Gene Expression Omnibus (GEO) (GSE75920; Buus et al.[31]). Robust multi-array average (RMA) values were calculated from CEL-files for 'core' probes using the 'oligo' R package[57]. Batch effects from the four independent batches were corrected by adjusting expression values using the ComBat function ('sva' R package)[58]. Differentially expressed probe sets between the CD24+CD73− and CD24+CD73+ stages were filtered by exhibiting a $\log_2$ fold change greater than 3 and BH adjusted $p$ value below 0.05 using parametric F-test. Differentially expressed probe sets were subsequently filtered by Gene Ontology terms related to cell surface expression (GO:0009986 or GO:0005886) and relative $\log_2$ expression greater than 7.

For gene products with commercially available antibodies, the protein expression and distribution among γδ thymocytes were analysed by flow cytometry. CD117, CD200 and CD371 were selected for further analysis based on the following criteria: adequate staining for flow cytometry allowing separation of a positive and negative population, ability to further divide the CD24+CD73− or CD24+CD73+ stages and having similar expression pattern within the Vγ1.1+ and the Vγ2+ subsets when divided into the CD24+CD73−, CD24+CD73+ and CD24−CD73+ stages.

**Flow cytometry**. Antibodies against mouse CD3 (145-2C11), CD4 (RM4-5), CD8 (53-6.7), CD5 (53-7.3), CD9 (MZ3), CD24 (M1/69), CD25 (PC61), CD27 (LG.3A10), CD44 (IM7), CD45RB (C363-16A), CD62L (MEL-14), CD73 (TY/11.8), CD117 (2B8), CD121a (JAMA-147), CD122 (TM-β1), CD127 (A7R34), CD154 (MR1), CD160 (eBioCNX46-3), CD186 (221002), CD192 (475301), CCR6 (29-2L17), CD199 (9B1), CD200 (OX-90), CD200R (OX-110), CD314 (C7), CD371 (5D3/CLEC12A), JAML (4E10), Ly-6C (HK1.4), TCRδ (GL-3), TCRβ (H57-597), TCRVγ1.1 (2.11), TCRVγ2 (UC3-10A6), TCRVδ6.3/2 (8F4H7B7), TCRVδ4 (GL2) and NK1.1 (PK136) were purchased from BD Biosciences (Franklin Lakes, NJ, USA), BioLegend (San Diego, CA, USA) or eBioscience (San Diego, CA, USA). See Supplementary Table 1 for full list of antibody conjugates. Fixable viability dye (eFluor780 or eFluor506; eBioscience, San Diego, CA, USA) or 7-AAD (BioLegend, San Diego, CA, USA) was used to exclude dead cells in the analysis.

Single-cell suspensions from thymus or inguinal lymph nodes (LN) were generated by disruption and straining through a 70-µm cell strainer with a rubber plunger from a 5-mL syringe. Thymic cell suspensions were enriched for γδ T cells by depleting them with anti-CD4 and anti-CD8 magnetic beads (Miltenyi Biotec). For all flow cytometry staining, antibodies were diluted in brilliant stain buffer (BD Biosciences, San Diego, CA, USA; cat. no. 563794) and stained for 30–40 min in the dark on ice. For intracellular cytokine staining, single-cell suspensions were stimulated with phorbol 12-myristate 13-acetate (PMA; 25 ng per mL) and ionomycin (625 ng per mL) in complete medium including monensin (2.08 µg per mL) for 4 h at 37 °C. Cells were stained for surface markers, fixed and permeabilised using BD Cytofix/Cytoperm (BD Biosciences; cat. no. 554722). Permeabilised cells were stained with anti-IL-17A (TC11-18H10), anti-IFN-γ (XMG1.2), anti-PLZF (9E12), anti-AHR (T49-550) and anti-RORγ(t) (B2D) or a rat IgG1 isotype control. All samples were analysed on a BD LSRFortessa or BD Aria II at the Core Facility for Flow Cytometry, Faculty of Health and Medical Sciences, University of Copenhagen. Data were analysed using the R or FlowJo software (Treestar, Ashland, OR, USA).

**Sorting γδ thymocytes**. γδ thymocytes from eight mice were isolated and stained for flow cytometry as described above. To sort the 14 different populations, thymocytes were initially sorted into four samples using a yield mask for best recovery: Vγ1.1+CD73−, Vγ1.1+CD73+, Vγ2+CD73− and Vγ2+CD73+ cells. Each of these four samples were subsequently sorted as A, B and C or D, E, F and G populations directly into TRI reagent (cat. no. T9424, Sigma-Aldrich) for RNA sequencing (RNA-Seq) or OP9 medium for co-culture experiments using a purity mask for optimal purity. To achieve sufficient cells for RNA-sequencing libraries, RNA was

pooled from two or three independent sorting experiments, each using thymi from eight mice. Due to the limited number of cells and direct lysis, post-sort purity was assayed only from the largest population ($V\gamma2^+$ B) and found to consistently exceed 95%. Cell lysates in TRI reagent were stored at −80 °C after final sorting until RNA extraction.

**mRNA sequencing.** Cell lysates in TRI reagent were thawed at room temperature, and RNA was extracted by adding BCP (cat. no. B9673, Sigma-Aldrich), centrifugation and using the upper phase. RNA was precipitated with isopropanol and washed twice with 70% ethanol. RNA samples were resuspended in nuclease-free water and stored at −80 °C until library construction. RNA-Seq libraries were constructed using the SMARTer Ultra Low Input RNA v4 (cDNA synthesis and amplification; cat. no. 634890, Takara) and SMARTer Low Input Library Prep Kit V2 (library prep from cDNA; cat. no. 634899, Takara) according to the protocols supplied by the vendor. Two biological replicates from each population were generated and sequenced during separate runs. For both independent biological replicates, sequencing was conducted as 100-bp paired-end reads divided on two lanes containing A to G populations from either the $V\gamma1.1^+$ or the $V\gamma2^+$ subset on the Illumina HiSeq2500 platform. This division on lanes assured high-fidelity comparisons of populations within each subset. The quality of the raw RNA-Seq reads was controlled using FastQC. Library preparation and sequencing were performed at AROS Applied Biotechnology A/S (Aarhus, Denmark).

**Statistics.** Unless stated otherwise, statistical analyses were performed using the two-sided Student's $t$ test with a 5% significance level, unpaired observations and equal variance. Significance was defined as *$p \leq 0.05$; **$p \leq 0.01$; ***$p \leq 0.001$.

**Bioinformatics analysis of flow cytometry data.** Equal numbers of $\gamma\delta$ T cells were down-sampled from each replicate, concatenated to a single fcs file using FlowJo and used for analysis in R using the 'flowCore' and 'cytofkit' packages[59, 60]. t-SNE was calculated for 10,000 cells based on the expression of CD24, CD25, CD73, CD117, CD200 and CD371. Due to the stochasticity of the algorithm, multiple t-SNE graphs were generated that showed a highly similar grouping. Cells were clustered using Phenograph, ClustalX, DensVM and FlowSOM[61]. FlowSOM clustering with $k = 7$ yielded similar-sized clusters with discernible differences in surface marker expression and was chosen for further analysis. The cluster profile heat map was created based on the mean expression of cells within each cluster. Progression analysis was conducted using the Isomap and Diffusionmap algorithms on a cluster-based down-sampled data set, as recommended in the cytofkit manual.

**Bioinformatics analysis of mRNA sequencing.** Raw RNA-Seq reads were pseudoaligned to the mouse transcriptome (ENSEMBL, GRCm38 release 79) using Kallisto. Expression analysis was conducted using the 'tximport' and 'DEseq2' R packages adhering to the guidelines suggested in the DEseq2 manual[62, 63]. Forty-one erythrocyte-associated transcripts were excluded due to high erythrocyte transcription within the V1C_2 sample in comparison to all the other samples (including its biological replicate V1C_1). The C populations from the second biological replicates (V1C_2 and V2C_2) clustered closer to their respective D populations compared with the first C population samples, likely due to the small number of cells and less efficient CD200 staining during sorting, leading to the inclusion of CD200$^{low-int}$ cells in the C populations of this replicate. To conservatively represent the expression within the populations, the mean of the biological replicates was used in data visualisations. Overall, transcriptional similarity was used to infer likely developmental relationships between each of the A to G populations. This tactic assumes that two populations exhibiting low overall transcriptional difference are more likely to be related than two populations exhibiting greater difference. Transcriptional differences between the populations were calculated using their pairwise Euclidian distances as well as by principal component analysis (PCA) calculated from all expressed RNA-Seq transcripts using R.

**TCR V$\delta$ usage and CDR3 clonotyping.** TCR sequence analysis was performed by aligning RNA-Seq sequences against TCR genes allowing insertions and gaps using miXCR software with settings optimised for RNA-Seq libraries including two assemblePartial and one extendAlignments runs, as recommended in the manual[64]. VDJtools[65] was used to eliminate erroneous clonotypes, filter for functionally recombined CDR3 sequences (by removing out-of-frame transcripts) and reduce the number of reads from each library to equal numbers (1200 and 724 aligned reads per library for CDR3$\delta$ and CDR3$\gamma$, respectively). The heat maps of enrichment of CDR3 motifs associated with different $\gamma\delta$ T cell effector subsets in Fig. 3h were extracted using 'regular-expressions' and quantified by dividing by the total number of filtered reads aligned to the TCR V$\delta$ or TCR V$\gamma$ locus in R. The heat map of TCR V$\delta$-chain usage in Fig. 4d within each population was calculated by the number of reads divided by the total number of filtered reads aligned to the TCR V$\delta$ locus in R. The CDR3 diversity quantification included in Supplementary Fig. 7 was calculated as the inverse Simpson index using the 'tcR' package in R[66] and using VDJtools.

**Code availability.** The code used for analysis in our study is available from the corresponding author on request.

**Data availability.** The RNA-Seq data sets used in Figs. 3 and 4 have been deposited in the GEO under ID code GSE97137. The previously published and publicly available MicroArray data set used in Fig. 1 is available in the GEO under ID code GSE75920. All other data are available from the authors on request.

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

## Acknowledgements

This study was supported by The Novo Nordisk Foundation (J.P.H.L.; Grant No. NNF10OC1013296). N.Ø. was supported by The Danish Cancer Society (Kræftens Bekæmpelse), the Fight Cancer Program (Knæk Cancer), the Novo Nordisk Research Foundation, The Novo Nordic Foundation Tandem Program, the Lundbeck Foundation and The Danish Council for Independent Research. C.G. was supported by The Danish Council for Independent Research | Medical Sciences and the Læge Sofus Carl Emil Friis og hustru Olga Doris Friis Foundation.

## Author contributions

T.B.B. and J.P.H.L. conceived the study. T.B.B. performed the experiments, analysed the data and wrote the original draft. T.B.B., J.P.H.L, N.Ø. and C.G. reviewed, edited and corrected the manuscript. J.P.H.L, C.G. and N.Ø. supervised the project, acquired the funding and provided expertise and feedback.

## Additional information

**Competing interests:** The authors declare no competing financial interests.

