## [Peer Review File · Nature Communications]

Reviewers' comments:

Reviewer #1 (gd17, gd thymocyte)(Remarks to the Author):

Buus and colleagues have clarified various issues and provided additional data to support their claims. They also re-qualified some interpretations that were somewhat dubious in the previous version of the manuscript. This notwithstanding, I remain skeptical of impact these new markers will have on the study of thymic gd T cell development, and I find the paper, by its own nature, too descriptive for an "article" in Nat Comm.

Nonetheless, it is undoubtedly a large amount of work that contains valid and potentially interesting information, even though the "new biology" is thin (restricted to some consideration of the role of the TCR in the developmental transitions that are proposed).

A major unresolved issue is acknowledged by the authors in their rebuttal, where they propose an appropriate solution:

"If found to be appropriate by you and the editor, we propose to conduct additional experiments showing the expression of CD122, CD27, NK1.1, CCR6, CD44 and CD45RB within the A through G populations. This would allow the reader to assess the relationship between the well established and newly presented markers as well as help establish how previous conclusions relate to our findings".

I certainly think such data would be very useful for the community - and facilitate the understanding of the reader used to pre-established markers other than the proposed in this study.

Reviewer #2 (gd T cell, ILC)(Remarks to the Author):

The authors have addressed all issues raised by the two reviewers in response to the original submission, however, mostly not experimentally. Indeed, they did a good job clarifying many points that were raised.

As it stands, the manuscript basically represents a very detailed and precise description of adult stages of gd T cell differentiation, which is per se a valuable contribution to the field. The novel aspect in this work is the inclusion of CD200 and CD371, markers which hitherto have not been used to stage steps of gd T cell development. While additional data on functional aspects of these molecules may be beyond the scope of this manuscript, the story would surely benefit from putting the stages proposed according to these markers in context with established markers. As suggested in their reply, it will be important to "conduct additional experiments showing the expression of CD27, CD44, CD122, CD45RB, CD62L, NK1.1, CD5, CD127, Ly6-C etc. within the A through G populations." and to present these new data in an intelligible way in a revised manuscript.

Reviewer #1:

Buus and colleagues have clarified various issues and provided additional data to support their claims. They also re-qualified some interpretations that were somewhat dubious in the previous version of the manuscript. This notwithstanding, I remain skeptical of impact these new markers will have on the study of thymic $\gamma\delta$ T cell development, and I find the paper, by its own nature, too descriptive for an "article" in Nat Comm.

We greatly appreciate your feedback that allowed us to clarify many of the issues in the manuscript. While the first parts of the study are descriptive the later parts does disrupt the steady state *in vivo* or *in vitro* to go beyond mere observations. Furthermore, we would argue that the descriptive part with the accompanying comprehensive total mRNA-seq dataset (with 14 different $\gamma\delta$ thymocyte populations each with 2 replicates) is in itself a valuable contribution to the field and a prerequisite for subsequent "interventive" experiments. We submit as an "article" to nature communications as they do not offer a "resource" format, and furthermore we would argue that the manuscript contains much data that goes beyond observations and provide new biological insights.

Nonetheless, it is undoubtedly a large amount of work that contains valid and potentially interesting information, even though the "new biology" is thin (restricted to some consideration of the role of the TCR in the developmental transitions that are proposed).

We are happy that the efforts put into this work shines through. As mentioned above, we agree that the initial part of the study is descriptive, but we would like to point out that in addition to the TCR stimulation data, the later sections of the manuscript also show "new biology" in the FTY720 treatment experiments elucidating three separate "terminal" thymic populations of $\gamma\delta$ T cells as well as indicating the distributions of the exported effector cells (which is different from their steady state proportion within the thymus).

A major unresolved issue is acknowledged by the authors in their rebuttal, where they propose an appropriate solution: "If found to be appropriate by you and the editor, we propose to conduct additional experiments showing the expression of CD122, CD27, NK1.1, CCR6, CD44 and CD45RB within the A through G populations. This would allow the reader to assess the relationship between the well established and newly presented markers as well as help establish how previous conclusions relate to our findings". I certainly think such data would be very useful for the community - and facilitate the understanding of the reader used to pre-established markers other than the proposed in this study.

We completely agree that providing a link between the previously established and newly presented markers would be very valuable for the $\gamma\delta$ community and the readers of the manuscript. As suggested, we have conducted these experiments and included histograms of each marker within each of the seven identified populations in Fig. 2h (and within the TCRV γ 1.1⁺ and TCRV γ 2⁺ subsets in Supplementary Fig. 2d,e). All markers were analyzed as PE conjugates (except CCR6-APC) on a flow cytometer with a yellow/green laser thus giving optimal resolution of expression. As mentioned in the previous response, despite their differential expression between some of the populations, these markers cannot clearly segregate all the newly identified populations presented in the study – particularly the C and D populations cannot be distinguished from the B population without CD200 and CD371 and the E and F populations cannot be segregated without CD117. Nonetheless the expression patterns of these markers supports the conclusions of our study and helps translate the findings and the resources provided in the manuscript to other labs using the previously established markers.

Reviewer #2:

The authors have addressed all issues raised by the two reviewers in response to the original submission, however, mostly not experimentally. Indeed, they did a good job clarifying many points that were raised.

We are happy that we could address the raised points and we appreciate your comments and suggestions which we believe have improved the manuscript.

As it stands, the manuscript basically represents a very detailed and precise description of adult stages of gd T cell differentiation, which is per se a valuable contribution to the field. The novel aspect in this work is the inclusion of CD200 and CD371, markers which hitherto have not been used to stage steps of gd T cell development. While additional data on functional aspects of these molecules may be beyond the scope of this manuscript, the story would surely benefit from putting the stages proposed according to these markers in context with established markers. As suggested in their reply, it will be important to “conduct additional experiments showing the expression of CD27, CD44, CD122, CD45RB, CD62L, NK1.1, CD5, CD127, Ly6-C etc. within the A through G populations.” and to present these new data in an intelligible way in a revised manuscript.

As mentioned in the reply to reviewer 1, we agree that this is an important addition to the manuscript which will facilitate easier integration with other studies using previously suggested markers. We have now analyzed the expression of all these markers as well as CCR6 within each of the 7 identified populations. We have visualized the expression of each of the markers within each population as histograms (Fig. 2h) as well as intensity colored t-SNE maps (Fig. 2i,j). As some of the existing markers exhibit subset specific expression patterns we have also included histograms for each of the seven populations divided into the TCRV γ 1.1⁺ and TCRV γ 2⁺ subsets (Supplementary Fig. 2d,e). We hope that these visualizations together provides an intelligible presentation of the expression (if not, we are very open for suggestions on how to represent this multi-dimensional expression data).

Reviewers' comments:

Reviewer #1 (Remarks to the Author):

The authors have performed the suggested experiment and added the results to the revised paper. While the data are valuable, they have abstained from discussing them properly (in the discussion of the paper) because they "believe this would divert attention from the conclusions of the study and may confuse the reader". Instead, I strongly feel that they MUST (in their own words) "further elaborate and include a more detailed discussion of the non-redundancy between the newly identified markers: CD117, CD200 and CD371, and the previously established markers". Of note, this had already been requested by the Reviewers' and, in especially, the Editor: "presented/discussed clearly to demonstrate the physiological relevance of the proposed gd thymocyte subsettings".

Furthermore, there is still a major inconsistency in the manuscript, which I have stressed in my comment to their original submission:

"There is great inconsistency and misinterpretation of the data regarding gdT17 cells throughout the paper, to the point that population G, acknowledged at multiple instances (see pages 8 and 17) as containing transcriptional and TCR repertoire signatures of gdT17 cells, is presented in the final model (Fig 7) as the gdNKT subset. In fact, it seems that population G is an heterogeneous mixture of gdNKT and gdT17 cells, as apparent in the transcriptional data of Fig 3g. This issue would likely be solved by employing NK1.1 and CCR6 as additional markers, as proposed already in 2009 by Prinz and colleagues (Haas et al. EJI 2009)".

The authors need to solve this issue – and should suggest the use of NK1.1 versus CCR6 to discriminate the two subsets within the G population. Moreover, what is their evidence that gd17 T cells (also) derive from population C? In fact, to add to the previous mRNA and TCR repertoire data, they now provide evidence that the typical gd17 T cell phenotype, i.e., CD44^{hi} CCR6⁺ CD27⁻, is contained in population G. This must be clarified in the text and in the (currently misleading) Figure 7 in a revised version of this manuscript.

Reviewer #2 (Remarks to the Author):

As suggested, the authors have performed additional experiments to relate their classification of adult gd T cell stages to previously described markers and presented the data for total gd T as well as for Vg1 and Vg2 separately. Thus, the MS should now be ready for publication.

Reviewers' comments:

Reviewer #1 (Remarks to the Author):

The authors have performed the suggested experiment and added the results to the revised paper. While the data are valuable, they have abstained from discussing them properly (in the discussion of the paper) because they “believe this would divert attention from the conclusions of the study and may confuse the reader”. Instead, I strongly feel that they MUST (in their own words) “further elaborate and include a more detailed discussion of the non-redundancy between the newly identified markers: CD117, CD200 and CD371, and the previously established markers”. Of note, this had already been requested by the Reviewers’ and, in especially, the Editor: “presented/discussed clearly to demonstrate the physiological relevance of the proposed gd thymocyte subsetings”.

We value your input and, as offered in the previous response, have gladly included a more detailed discussion of the non-redundancy between the existing and new markers in relation to the previous established markers (p. 14). In terms of the physiological relevance, this is what the Figures 3 through Figure 6 all investigate and show that these markers segregate three development pathways leading to the development of distinct $\gamma\delta$ T cell effector subsets exhibiting hallmarks of $\gamma\delta$ T1, $\gamma\delta$ NKT and $\gamma\delta$ Tn cells. As existing markers cannot distinguish the C+D and E populations from the B and F populations, respectively, such segregation into functional pathways would not be possible without the addition of new markers. We are not sure what additional demonstrations of physiological relevance you would like us to discuss.

Furthermore, there is still a major inconsistency in the manuscript, which I have stressed in my comment to their original submission:

“There is great inconsistency and misinterpretation of the data regarding gdT17 cells throughout the paper, to the point that population G, acknowledged at multiple instances (see pages 8 and 17) as containing transcriptional and TCR repertoire signatures of gdT17 cells, is presented in the final model (Fig 7) as the gdNKT subset. In fact, it seems that population G is an heterogeneous mixture of gdNKT and gdT17 cells, as apparent in the transcriptional data of Fig 3g. This issue would likely be solved by employing NK1.1 and CCR6 as additional markers, as proposed already in 2009 by Prinz and colleagues (Haas et al. EJI 2009)”.

The authors need to solve this issue – and should suggest the use of NK1.1 versus CCR6 to discriminate the two subsets within the G population.

You are absolutely right. While we in our previous responses clarified the reasons for why we assign the G population to be the thymic precursors of gdNKT cells in the adult thymus, we did not include a comprehensive argument in the discussion of the manuscript. We have corrected this and included a detailed discussion for why we consider the gdT17 cells included in the G population as being long-lived resident effector cells reminiscent of fetal thymic T cell development (p. 16). We have also included the recommendation for using the effector markers CCR6 and NK1.1 to further discriminate the two subsets.

Moreover, what is their evidence that gd17 T cells (also) derive from population C? In fact, to add to the previous mRNA and TCR repertoire data, they now provide evidence that the typical gd17 T cell phenotype, i.e., CD44^{hi} CCR6⁺ CD27⁻, is contained in population G. This must be clarified in the text and in the (currently misleading) Figure 7 in a revised version of this manuscript.

This is a good point and we agree that the placement of gdT17 cells in figure 7 can be confusing. We fully agree that gdT17 cells can be found within the G population at steady state, but as previously mentioned, gdT17 cells do not develop in the adult thymus and are unlikely to be derived from the "F" population (as this population express many markers associated with having experienced strong TCR signals). We have hinted that gdT17 cells may be derived from the "C" population in the fetal thymus based on the "C" populations lack of TCR selection and its expression of transcription factor networks tightly associated with gdT17 cells (as shown by Malhotra et al. 2013). However, we agree that the evidence for this is circumstantial and should not be hinted in the figure without more direct experimental evidence. We have revised Figure 7 and the corresponding legend to avoid this confusion and indicated how CCR6 and NK1.1 can be used to discriminate the resident effectors included in population G. We believe that now the revised summary figure 7 better reflects the findings of the study.

Reviewer #2 (Remarks to the Author):

As suggested, the authors have performed additional experiments to relate their classification of adult gd T cell stages to previously described markers and presented the data for total gd T as well as for Vg1 and Vg2 separately. Thus, the MS should now be ready for publication.

REVIEWERS' COMMENTS:

Reviewer #1 (Remarks to the Author):

The paper is now suitable for publication.